# Maize variety preferences among smallholder farmers in Ethiopia: Implications for demand-led breeding and seed sector development

Paswel Marenya[1]*, Rosina Wanyama[2], Solomon Alemu[3], Ola Westengen[4], Moti Jaleta[5]

1 Sustainable Agrifood Systems Program, International Maize and Wheat Improvement Center (CIMMYT), Nairobi, Kenya, 2 Enabling Impact Flagship, World Vegetable Center, Arusha, Tanzania, 3 Standing Panel on Impact Assessment, CIAT-Bioversity Alliance, Addis Ababa, Ethiopia, 4 Faculty of Landscape and Society, Norwegian University of Life Sciences NMBU, Ås, Norway, 5 Sustainable Agrifood Systems Program, CIMMYT, Addis Ababa, Ethiopia

* p.marenya@cgiar.org

**Data Availability Statement:** All relevant data are within the paper.

## Abstract

Among smallholder maize farmers in Ethiopia (and similar areas in Africa), yield and stress tolerance traits in maize varieties are important. While high yields remain a major objective, breeding and seed system development programs are increasingly based on the recognition that farmers also have an interest in other agronomic and consumption traits. In this paper we illustrate these issues by measuring the trade-offs farmers may be willing to make for specific traits in the mid-altitude maize markets in Ethiopia. Based on Choice Experiments among 1499 respondents, we estimate the preference for a set of agronomic and consumption traits relative to yield. by capturing farmers' "willingness to sacrifice yield". The results suggest a significant willingness to sacrifice yield for drought tolerance among both male and female household members, but not for early maturity *per se*. There was also a high willingness to sacrifice yields for plant architecture traits like closed tip and lodging resistance among male participants, but not among females. Heterogeneity in responses according to gender, education and land area under maize cultivation suggests that market segmentation is necessary for seed system development to become more demand-led and inclusive. Final and realistic segmentation will depend on the commercial viability or social impact potential of each segment.

## Introduction

Like much of eastern and southern Africa, maize has acquired a prominent role in Ethiopian diets and in terms of total national annual production it outstrips that of *teff*, sorghum and wheat by more than 50% each [1]. Maize provides a cheaper calorie source compared to the other cereals and now makes up the largest share in calorie intake and production in the country [2–4]. The high yields of maize compared to the other cereals is a general pattern seen around the world and it is partly due to the investment by public and private actors in maize breeding. The maize breeding system in Ethiopia goes back more than five decades, with

**Funding:** PM: Bill and Melinda Gates Foundation, Grant INV-003439 and OPP1134248 https://www.gatesfoundation.org/ OW and MJ: Research Council of Norway, Grant RCN 288493 https://www.forskningsradet.no/en/.

**Competing interests:** The authors have declared that no competing interests exist.

increased investment in the last two decades [2, 5]. The growing importance of maize in Ethiopia's agriculture has, however, been based on just a few market-dominant varieties.

## Ethiopia's maize breeding priorities

The review by [6] shows that Ethiopia's maize breeding program started in early 1950s (about 70 years ago). The national breeding system is currently organized around four major agroecological zones. These include mid-altitude sub-humid, highland subhumid, low-land subhumid and low moisture agroecologies [6]. These agroecologies are characterized by differences in altitude, annual rainfall, temperature, relative humidity and latitude. The emphasis on these decades long program has been to provide a wide range of locally adapted open pollinated varieties (OPVs) especially in the early years and later hybrids have become common. In recent years, emphasis has been placed on tolerance to abiotic stresses such as tolerance to foliar diseases such as grey leaf spot, leaf blight, common leaf rust, striga weed (*Striga hermonthica*) and the like. Since the 2000s, breeding programs for nutritional quality have also been implemented with efforts made to breed for grains with high protein, provitamin-A as well as good livestock feed attributes. The implementation of participatory variety selection methods (where farmers comparatively score pre-released varieties for various attributes) have been implemented to ensure new released maize varieties meet end user preferences. The geographically dispersed breeding programs have generated varieties in different maturity classes such as extra early OPVs for drought escape which mature at 90 days for the drought prone areas. Intermediate maturing varieties are targeted at the cooler mid- to high altitudes. In terms of appealing to different types of farmers with varying economic capacity to purchase inputs, the breeding programs have taken the combined approach of developing varieties that respond to high input levels but also are tolerant to low fertilizer application among most resource-poor farmers.

Between 2002–2010, over 90% of total maize seed sales were accounted for by three varieties produced mainly by the Ethiopian Seed Enterprise (ESE). One of these leading varieties was BH660, which was reported to have approximately 50% market share of hybrid seed sales in Ethiopia (Worku et al. 2012). In recent years more varieties with emphasis on drought tolerance (and tolerance to foliar diseases) have also been developed and are now gaining market share. By 2014, a total of 61 different maize varieties were developed over the years [2]. The authors in [2] counted a total of 20 different commercial varieties (hybrids and composites) as of 2013, BH660 had 51% market share. Recent government of Ethiopia reports indicate that more than 68 different maize varieties released in Ethiopia in more recent years [7]. Therefore, it is apparent the country is on a path towards creating a pipeline of new generation hybrids suitable for high, mid, and low altitude zones. This is an indication of progress in Ethiopian seed systems to provide a more diverse varietal portfolio. A better evidence base about farmers' trait preferences is a key dimension for ensuring demand-oriented maize seed system development going forward [8, 9]. While yield obviously is an important characteristic for maize varieties, there is also a range of other agronomic and end-use traits that warrants more attention from breeders and other seed sector actors. For a more detailed analysis of Ethiopia's maize breeding programs in the last seventy years, please refer to [6, 10].

In this paper, we use an innovative approach to measure farmers' varietal attribute preference by estimating their willingness to sacrifice yield (WTSY) (for more on the WTSY concept, see [11]), where WTSY was first estimated for a similar study in western Kenya. Estimating WTSY can indicate the relative trade-offs farmers are willing to make between traits.

Previous studies from other smallholder contexts have shown that trait preferences can explain why farmers continue to cultivate different maize varieties. The authors in [12] working within a Malawian context, found that while farmers indeed appreciated hybrids for their

yield and drought tolerance traits, they rated local OPVs higher for storability and some consumption traits and the authors suggested this as a potential explanation for the 30–40% adoption plateau for hybrid maize in the country [12]. A series of papers on perceived benefits of different types of maize varieties in Mexico have similarly shown that farmers plant hybrids for yield, they continue to grow landraces as well because they find them superior for other agronomic and processing traits [13–15]. Plant breeders have employed different strategies for meeting the diverse demands of farmers, ranging from various forms of collaborative plant breeding to market-based approaches to product profiling [16]. However, as climate, pest- and disease-pressure and farmer preferences constantly change, plant breeding and seed systems must not only deliver a more diverse portfolio of varieties, but also enable a higher varietal turnover than what is currently the case in the region [17, 18].

Under some circumstances, stress tolerance and yield attributes could outweigh common "taste attributes" if the varieties lack preferred grain characteristics but are high yielding and stress tolerant. Hypothetically, new tastes can be acquired, or less preferred tastes can be overcome, if the choice boils down to sacrificing yield and stress tolerance traits. Arguably, farmers' knowledge about the positive attributes of new varieties and on what specific attributes they outperform the existing ones also helps build demand for new varieties. We illustrate these concepts in this paper.

The rest of this paper is organised as follows. The next section describes the data collection and empirical methodologies used in the study. The methodologies section has subsections that discuss the sampling and data sources, the choice experiment design, and the underlying choice theory used in this study. The methodology section also describes the traits used in this study and outlines the econometric approaches. The results section presents the descriptive data, the mixed logit model results and finally the WTSY as measured by the choice experiment (CE) and the BDM data. The final section concludes by summarising key results and describes the implications therefrom.

## Materials and methods

### Sampling and data

The data used for this study came from a random sample of 800 households in the mid-altitude sub-humid maize growing zones of western and central agro-ecozones of Ethiopia (Table 1). The sampling process was done in three stages by combining purposive and random sampling. In the first stage, districts (*woredas*) that were in the mid-altitude maize growing areas of Ethiopia were selected. These areas coincided with a maize breeding and seed systems development projects (the Stress Tolerant Maize for Africa Project and later the Accelerating Genetic

**Table 1. Sampled districts, villages and households.**

| Region | Zone | *Woreda (District)* | Number of villages | Number of households |
|---|---|---|---|---|
| Oromiya | Arsi-zone | Zeway Dugda | 2 | 80 |
| | East Shewa-zone | Adama zuriya | 2 | 80 |
| | | Dugda | 2 | 80 |
| | West Arsi-zone | Arsi Negele | 3 | 120 |
| | | Shala | 2 | 80 |
| | | Shashemene zuria | 4 | 160 |
| | | Siraro | 2 | 80 |
| SNNP | Siltie-zone | Alaba special | 3 | 120 |
| **Total** | **4** | **8** | **20** | **800** |

Gains in Maize and Wheat Project co-implemented by CIMMYT and Ethiopian researchers) with the former project having been implemented in these districts between 2015–2020 and the latter project ongoing as at the time of writing.

The second stage involved the selection of villages (*kebeles*) within the project sites. Villages were selected using a sampling design that makes explicit use of the population size, "the probability proportional to size" (PPS) in order to sample more villages (and consequently households) from the more populous *woredas*. In the third stage, a random sampling of households within each village was selected from a village household listing developed by the village leader and the project team. Using a random number generator, 40 households were randomly selected from each village. Based on this procedure, 800 households from two regions Oromia and Southern Nations, Nationalities and Peoples region (one of the nine federal administrative regions of Ethiopia–the latter being SNNP for short), four zones, eight *woredas* and 20 *kebeles* (villages) were selected (Table 1). In each household, both the household-head and the spouse were interviewed. In cases where a spouse was not available, any adult household member that was knowledgeable about farming in the household was interviewed (in addition to the household head).

## Focus group discussions

Prior to the experiments, extensive focus group discussions were held with farmers in the survey communities. Key traits identified during the discussions included: yield, drought tolerance, maturity period, lodging resistance, husk (tip cover), resistance to foliar diseases, and taste when boiled or roasted. A summary of key traits identified and their distribution across the four experiments is presented in Table 3. All the traits were used for both CEs and BDM. More information on the BDM and the CE processes are presented in Appendices 1 and 2 in S1 File respectively.

## Choice experiment design to elicit maize trait preferences

A CE was conducted to elicit farmers' preferences for selected yield and non-yield maize traits. While CEs have been applied in diverse fields such as transportation [19], health [20], marketing [21], and environmental economics [22], they have more recently found application in studying agricultural value chains such as consumer attitudes and preferences for nutrient-dense staples [23], farmers valuation of agronomic traits [24], and credence attributes [25]. Other examples are preferences for alternative marketing channels and supply chain differentiation [26, 27], and farmers' valuation of alternative input policy proposals [28].

The CE techniques are stated experiments where individuals are asked to choose their preferred alternatives from sets of abstract product profiles ('choice sets'). The concept is motivated by Lancaster's consumer choice theory: consumers derive utility from the underlying characteristics or attributes of a service or product [29] based on the random utility theory (RUT). In RUT, rational consumers prefer an alternative that yields the highest subjective utility among a given set of alternatives [30–32]. Some of the factors that drive utility can be observed by the analyst. These observed factors form part of the deterministic (systematic) part of the utility function. Other factors are not observable (e.g., unobserved alternative attributes; individual characteristics, also called 'unobserved taste variations) and others are due to measurement or specification errors [33, 34]. The basic theory is that utility is not directly observable, but can be deduced from farmers' choices (here, 'stated preferences').

Following this framework, we proceeded as follows. Farmers were presented with two hypothetical maize varieties exhibiting varying trait combinations. For the $n^{th}$ farmer faced with $J = 2$ varieties, the utility of variety $j$ is $U_{nj}$. By choosing variety $j$, the implication is that

$U_{nj}$ is the maximum among the $J$ utilities. As we explain below the estimation of the preference weights is based on a statistical model driven by the probability that variety $j$ is chosen is;

$$Prob\left(u_{nj} > u_{nk}\right) \quad \text{for all } j \neq k; k = 1, 2 \ldots J$$

## Data collection

The data collection was done using a structured questionnaire that had one section dedicated to the CE and additional sections with survey questions that captured demographic and farm information. The CE elicited maize trait preferences from participants in ways that required them to give up some levels of one trait in favor of another trait, thereby improving on other stated preference approaches by compelling participants to make trade-offs. To generate the CE sessions, we used the D-efficient statistical design. The traits and trait levels used for the CEs are described in Table 3 below.

Each household was randomly assigned to one of the four experiments using an excel number generator, (resulting in 200 households in each experiment). Table 2 shows the distribution of respondents interviewed for each experiment. In each household, efforts were made to elicit the choices from both the household head and the spouse. However, the actual interviews were conducted separately with the household head and the spouse (or another adult household member who independently operated a maize plot). In cases where there was only one single adult in the household (with only minor children as the other members) or where only one adult operated a maize plot, then only that one member (typically the household head) was interviewed. From this process, a total of 1499 respondents participated in the four experiments (Table 3).

## Econometric and estimation strategies

In basic terms, data from CE models tend to be analyzed using multinomial logit (MNL) models. The MNL (and related conditional logit–CL) models have been widely used to analyze discrete choice problems [35–37]. The MNL has the individual as a unit of analysis and uses the individual's characteristics as arguments in the estimation model, while CL focusses on the set of alternatives for each individual and the characteristics of those alternatives used as explanatory variables [37]

The two models can be specified as follows

$$P_{nj} = \exp\left(X_n\beta_j\right)/\sum_{k=1}^{J}\exp(X_n\beta_k)\text{for multinomial logit, and} \qquad (1)$$

$$P_{nj} = \exp\left(Z_{nj}\delta\right)/\sum_{k=1}^{J}\exp(Z_{nk}\delta)\text{ for conditional logit} \qquad (2)$$

**Table 2. Distribution of respondents across experiment groups.**

| Experiment | Respondent categories | | | | Sex of all respondents | | |
|---|---|---|---|---|---|---|---|
| | HH head | Spouse | Other adult | Total | Male | Female | Total |
| Choice experiment A | 201 | 161 | 13 | 375 | 186 | 189 | 375 |
| Choice experiment B | 197 | 163 | 7 | 367 | 181 | 186 | 367 |
| Choice experiment C | 201 | 173 | 12 | 386 | 195 | 191 | 386 |
| BDM Auctions | 199 | 160 | 12 | 371 | 190 | 181 | 371 |
| | 798 | 657 | 44 | **1499** | 752 | 747 | **1499** |

**Table 3. Summary of maize traits identified and used in the experiments.**

| Experiment | Traits† | Description | Levels | Reference |
|---|---|---|---|---|
| **A, B, C** | Yield (*Quintals/timad*) | In Ethiopia yield is measured in Quintals/*timad*. A quintal is a **100kg** bag and a *timad* is quarter of a hectare. | 2, 4, 6, 8, 10 | 2 |
| **A** | Drought tolerant | When there is a dry spell, the tolerant maize stays green. The variety which is moderately or drought tolerant has the quality of yielding at least half of the normal yields when there is mid-season moderate (non-catastrophic) drought. | Not tolerant, moderately tolerant, completely drought tolerant | Not drought tolerant |
| | Maturity period | The time from planting to when the maize has filled up the grains and can be harvested for roasting or cooking. Meaning that harvesting is imminent, and moisture stress is no longer a threat to maize yield. | Matures in 3 months, Matures in 4 months | Matures in 4 months |
| **B** | Lodging resistance | This is what happens when the maize cannot stand and easily breaks and falls especially if there is windstorm. This can be devastating if it happens mid-season before grain sets and farmers cannot recover anything from the fallen plants. | Not resistant, moderately resistant, completely resistant | Not lodging resistant |
| | Husk (tip) cover | When the tip of the maize cob is open or is closed. The varieties with open tips are susceptible to grain rotting because they easily let moisture into the cob and leads to fungal infection and rotting. This can be a huge pre-harvest loss. | Open tip, Closed tip | Open tip |
| **C** | Resistance to foliar or other disease | Foliar disease occurs when there is some darkish diseased leaves and pests on them at varying degrees. This leads to diseased plants and low yields. | Not resistant, moderately resistant, completely resistant | Not resistant rust resistant |
| | Taste when roasted or boiled | This primarily refers to the sugary (sweet) taste of the maize gain when roasted or boiled. Some varieties are known for this and are thus preferred. When the grain is dry, the sugary taste disappears as most of the carbohydrates are complex starches. | Bland/not sweet when roasted, Sweet when boiled or roasted | Not sweet |

†All the traits were used also in the BDM auctions.

where $X_n$ are the demographics of the choice maker $n$, $Z_{nj}$ are the characteristics of alternative $j$ for individual $n$, while $\beta$ and $\delta$ are the corresponding vectors of parameters that represent the influence of individual and attribute characteristics on the choice.

To reduce the complexity of the choice experiments, we designed three separate experiments consisting of three traits each (with yield being common to all three experiments). For a subset (approximately 25%) of the experiments, we also conducted a Becker-DeGroote-Merschack (BDM) auction experiment [38]. The BDM element was added as a robustness check against hypothetical bias in the CEs. Using incentive-compatible mechanisms like the Vickrey's random $n^{\text{th}}$ price auction, or Becker-DeGroote-Marschack (BDM) framework [38] can reduce such biases [39].

Generally, the specification of MNL and CL models require that the unobserved effects are independently and identically distributed (IID) across the alternatives in the choice set, according to the extreme type 1 distribution. This means that the odds ratio between two alternatives does not change by the inclusion or exclusion of any other alternative [40]. For instance, assuming two choice sets, $C_1$ and $C_2$ such that $C_1 \subseteq C_n$ and $C_2 \subseteq C_n$, and for any alternatives $j$ and $k$ in both $C_1$ and $C_2$, the odds of choosing alternative $j$ over alternative $k$ should be independent of the choice set for all pairs $j, k$ or by the inclusion or exclusion of any other alternative;

$$\frac{P(j|C_1)}{P(k|C_1)} = \frac{P(j|C_2)}{P(k|C_2)} \tag{3}$$

The IID assumption results into a more rigid property of 'independence from irrelevant alternatives' (IIA) [34, 37]. The IIA property assumes that everybody in the population has a homogeneous preference structure, and therefore restricts the $\beta's$ to be the same for all

members of the population [41]. That is, $\varepsilon_{nj}$ for all $n, j$ the probability that a given individual $n$ chooses alternative $j$ within the choice set $C_n$ is given by;

$$p(j|C_n) = \frac{\exp(\mu v_n)}{\sum_{k \varepsilon c_n} \exp(\mu v_k)} \tag{4}$$

Therefore, the IIA property is a rather strong assumption to be satisfied by any method when there are only two alternatives, and it will be of minor importance with a few alternatives and a fixed or lightly-varying choice set [42]. The IIA can be remedied somewhat by assuming a multivariate normal distribution (allowing the residuals across alternatives to be correlated with each other) and estimating the model with multinomial probit model [40]. However, this approach has its drawbacks due to difficulties in estimation.

The alternative (which we employ in this paper) is to use the mixed logit model (MIXL) (also known as the random parameters model)–currently a more common approach in estimating choice data [19, 43, 44]. The MIXL obviates the three limitations of the standard logit models by allowing for random taste variations, unrestricted substitution patterns, and correlation in unobserved factors which relaxes the IIA assumption [45, 46]. In the MIXL, the utility derived by farmer $n$ from choosing maize variety $j$ on choice occasion $t$ is given by:

$$U_{njt} = (\beta + e_n)x_{njt} + \varepsilon_{njt} \quad e = 1, \ldots . E; j = 1, \ldots J; t = 1, \ldots T \tag{5}$$

where $\beta$ is the vector of mean attribute utility weights in the population and $e_n$ is the vector of person $n$'s specific deviation from the mean. The random error term $\varepsilon_{njt}$ is still assumed to be independently and identically distributed extreme value. The $e_n$ can be specified to take any distribution; normal, log-normal or triangular [36]. Although most applications use the multivariate normal, MVN (0, Σ), the price coefficient is sometimes assumed to be log-normal to impose positive sign restriction [47]. In this paper however, we replace the price variable with yield.

## Yield as the numeraire maize trait

If (as we here assume) most farmers want to maximize yield, all else equal, we estimate the WTSY, or the yield penalty they are willing to accept in exchange for a desirable trait. However, this is an experimental device, not a policy prescription because in real world situations, yield penalties are not accepted for new varieties, and neither is it in farmers interests for a variety to underperform in yield terms, compared to pre-existing varieties.

This is analogous (in our case) to the 'willingness to pay' or the cost (price) variable. In the absence of correlation between variety attributes, the MIXL model therefore takes the following form;

$$Y_{njt} = \gamma Q_{njt} + \delta Z_{njt} + \varepsilon_{njt} \tag{6}$$

where $Y$ is a binary decision variable that takes the value of 1 if farmer $n$ chooses variety $j$ in choice scenario $t$, and 0 otherwise. The variable $Q$ is the yield attribute–which was used in the place of commonly used price variable, while $Z$ is a vector of other non-yield maize traits. Recall from Table 3, the non-yield attributes were *drought tolerance*, *90-day maturity*, *resistance to lodging*, *husk/tip cover*, *resistance to rust/pest*, and *taste*. A positive coefficient for γ and δ implies a positive influence of yield and non-yield traits on selection of a particular variety. Estimation of Eq 6 gives the mean of the coefficient, and the standard deviation of the distribution of the coefficient around the mean. Preference heterogeneity is deemed present if the standard deviation is significant. We therefore extend Eq 6 by including interaction terms to better

understand the role of socioeconomic factors in influencing farmers' preferences.

$$Y_{njt} = \gamma Q_{njt} + \delta Z_{njt} + \lambda \left( \boldsymbol{Z_{njt}} * \boldsymbol{x_n} \right) + \varepsilon_{njt} \tag{7}$$

In Eq 7, $x$ is a vector of socioeconomic characteristics including age and education level of the respondent, and size of land under maize. Estimation of Eqs 6 and 7 follows the simulated maximum likelihood method as described by Hole (2007). It can be expected that in Eq 7 the responses are correlated which could mean that the intra-household responses are not independent. This is a fair assumption given that the respondents are part of the same household. To handle this, we implemented the MIXL in STATA 16. The estimation in this procedure was clustered at household level with robust standard errors. Note that the multiple observations within the household has a "panel" structure. In our case, the MIXL is designed to take this into account via and sub-routine "group_id" during MIXL implementation. The "group_id" was meant to identify the cluster of each respondent through the household identifier (household cluster), respondent type (intra-household respondent heterogeneity) and experiment session number (to capture any correlations between experimental sessions).

The estimates obtained from Eqs 6 and 7 can further be used to compute the willingness to pay for the selected attributes. Here, we compute farmers' WTSY for other preferred non-yield traits. We estimate the WTSY in a manner analogous to WTP, by obtaining the partial derivatives of yield ($Q$) with respect to other attributes ($z$), and multiplying by -1 [20];

$$'WTSY' = \frac{\partial Q}{\partial Z_j} = -\frac{\delta_j}{\gamma}$$

## Maize traits in relation to grain and seed prices

The use of yield as the "price" for evaluating maize traits is intuitive [11]. Consider the fact that despite the various stress tolerance, grain quality or agronomic traits that a maize variety possesses, maize grain itself is marketed as an undifferentiated product without any price differentials based on many of the field traits [48]. Although official grain quality standards and definitions do exist, these apply to international trade and meant to conform to phytosanitary and health standards in quality-sensitive international markets. In local domestic markets, other than obvious aspects such damaged or rotten grain or foreign matter or debris; there are no price differentials based on grain grades *per se* (e.g., uniformity, flintiness, size, color etc.) [48]. This means that although there is a drive towards market-driven maize breeding, unless producers are willing to pay more for newer maize varieties and a segmentation of the market is enabled, many seed companies may find only weak incentives to regularly update their maize variety portfolio, given the high R&D and marketing costs involved in the new varieties. This is evident in the frequently-reported slow variety turnover often cited in the region.

The seed markets in Ethiopia are similarly un-differentiated. Maize seed is mostly marketed by government-owned seed enterprises operated by federal or state governments. Though there is a seed price difference between varieties supplied by private seed producers (mainly Pioneer/CORTEVA) and the public seed enterprises, in each supplier group, seed prices do not vary based on a variety's attributes and tend to be controlled. Seed price is therefore not an attribute that reflects the value of a specific trait since seed prices tend to vary very little for different varieties. Confirming this, in focus group discussions (FGDs), seed price was not mentioned as an attribute that farmers considered when deciding on their seed purchases. This is understandable given the dominant role state enterprises play in these seed markets, a policy

choice driven by need for low-priced seed. For these reasons, price was not included as a cost variable, rather we use yield as the key consideration farmers make when choosing different varieties. For instance, when a farmer chooses an early maturing versus a late maturing variety the calculation boils down to the yield difference as the "price" a farmer would be willing to pay for a lower yielding, but early maturing variety compared to a higher yielding late maturing type.

## Results and discussion

### Demographic characteristics

The sample was relatively balanced between men and women, being about 50% for each category. The percentage of households who were maize net buyers was 29–31%. The average age of the respondents was 54.8 years. The data shows that the households were largely agrarian with 65% relying mainly on primary agriculture as the main farming occupation. There is some indication that most of the households are far from markets with an average of 11 km to the nearest trading center. The overall picture is that of a middle-aged rural farming population, dependent on agriculture and more than 80% with only primary level (eight years or less) of education (Table 4).

### Experimental results

In this section we review the CE results from MIXL and BDM models to identify the possible tradeoffs in farmers choice of maize traits in the study areas. To reiterate for emphasis, we use yield as the *numeraire* for comparing the WTSY for the other six traits (*drought tolerance, maturity in three months or less, lodging resistant, closed tip, sweet taste and resistant to foliar diseases*). In Experiment A (Table 5), the yield attribute had a statistically significant positive coefficient but half of that of drought tolerance in magnitude. The interaction effect suggests that there was heterogeneity with regards to education and land area under maize. These suggest that larger maize producers emphasized yield more than smaller maize producers. Had the FGDs not revealed the reason behind the issues, it could be seen as surprising that early

**Table 4. Sample characteristics.**

| Variable | Description | Experiment | | | |
|---|---|---|---|---|---|
| | | **A** | **B** | **C** | **BDM** |
| Respondent type | Household Head (%) | 53.60 | 53.68 | 52.07 | 53.64 |
| | Spouse (%) | 42.93 | 44.41 | 44.82 | 43.13 |
| | Another adult member (%) | 3.47 | 1.91 | 3.11 | 3.23 |
| Sex | = 1 if respondent is male (%) | 49.60 | 49.32 | 50.52 | 51.21 |
| Age | = 1 if respondent is below 35 years | 56.80 | 53.68 | 54.15 | 59.30 |
| Education | = 1 if respondent has post primary education (%) | 11.73 | 14.17 | 12.18 | 16.17 |
| Total land owned (*timad*) | *Timad* | 4.71 | 5.60 | 5.54 | 5.17 |
| Land under maize (*timad*) | *Timad* | 2.37 | 2.48 | 2.79 | 2.50 |
| Maize net buyer | = 1 if respondent is a net maize buyer (%) | 31.20 | 28.61 | 31.09 | - |
| Occupation | = 1 if main occupation of respondent is agriculture (%) | 65.33 | 66.21 | 64.77 | 66.85 |
| Dis. to nearest trading center | Km | 12.39 | 10.89 | 10.33 | 12.36 |
| Region | Oromia (%) | 86.93 | 82.29 | 84.46 | 85.71 |
| | SNNP (%) | 13.07 | 17.71 | 15.54 | 14.29 |
| *Number of households* | | *201* | *197* | *201* | *199* |
| *Number of respondents (N)* | | *375* | *367* | *386* | *371* |

**Table 5. Mixed logit results for Ethiopia with interaction terms: Experiment A (drought tolerance, maturity in three months or less) B (lodging resistant, closed tip) and C (sweet taste and resistant to foliar diseases).**

| Variables | Pooled | | Male respondent | | Female respondent | |
|---|---|---|---|---|---|---|
| | Coeff | SD | Coeff | SD | Coeff | SD |
| **EXPERIMENT A** | | | | | | |
| Yield | 1.166*** | | 1.066*** | | 1.362*** | |
| | (0.185) | | (0.269) | | (0.271) | |
| Drought tolerant | 3.662*** | 0.078 | 3.472*** | 0.169 | 4.162*** | 0.177 |
| | (0.488) | (0.217) | (0.739) | (0.205) | (0.697) | (0.233) |
| Matures in 3 months or less | -0.005 | 0.036 | -0.183 | 0.186 | 0.205 | 0.068 |
| | (0.211) | (0.114) | (0.331) | (0.148) | (0.289) | (0.125) |
| Yield*Age | -0.061 | -0.128 | 0.100 | -0.108 | -0.325 | 0.050 |
| | (0.151) | (0.122) | (0.213) | (0.148) | (0.225) | (0.099) |
| Yield*Education | 0.047** | 0.028* | 0.014 | -0.016 | 0.156*** | 0.057*** |
| | (0.024) | (0.016) | (0.029) | (0.017) | (0.054) | (0.022) |
| Yield*Land under maize | 0.220*** | 0.006 | 0.287*** | 0.015 | 0.168* | -0.008 |
| | (0.069) | (0.018) | (0.108) | (0.028) | (0.091) | (0.023) |
| Drought tolerant*Age | -0.048 | 0.204 | 0.293 | -0.222 | -0.638 | 0.080 |
| | (0.344) | (0.211) | (0.494) | (0.276) | (0.505) | (0.264) |
| Drought tolerant*Education | 0.111** | -0.074** | 0.043 | -0.029 | 0.337*** | -0.080* |
| | (0.054) | (0.030) | (0.068) | (0.026) | (0.119) | (0.046) |
| Drought tolerant*Land under maize | 0.544*** | -0.018 | 0.681*** | 0.020 | 0.433** | 0.046 |
| | (0.150) | (0.039) | (0.234) | (0.051) | (0.201) | (0.052) |
| Drought tolerant*Region[a] | -0.208 | 0.603*** | -0.163 | 0.671*** | -0.264 | 0.610*** |
| | (0.239) | (0.080) | (0.370) | (0.116) | (0.317) | (0.102) |
| Maturity*Age | -0.046 | 0.364*** | 0.080 | -0.461*** | -0.162 | 0.335*** |
| | (0.104) | (0.101) | (0.159) | (0.161) | (0.147) | (0.129) |
| Maturity*Education | -0.019 | -0.006 | -0.003 | 0.051*** | -0.046 | 0.004 |
| | (0.016) | (0.021) | (0.023) | (0.018) | (0.029) | (0.036) |
| Maturity*Land under maize | -0.085** | -0.015 | -0.071 | 0.001 | -0.094* | -0.004 |
| | (0.038) | (0.032) | (0.059) | (0.042) | (0.054) | (0.048) |
| Maturity*Region | -0.212 | -0.106 | -0.203 | 0.050 | -0.277 | -0.015 |
| | (0.148) | (0.089) | (0.229) | (0.101) | (0.200) | (0.083) |
| Observations | 9,000 | 9,000 | 4,464 | 4,464 | 4,536 | 4,536 |
| chi2 | 95.07 | 95.07 | 54.78 | 54.78 | 57.29 | 57.29 |
| Log Likelihood | -1850 | -1850 | -902.1 | -902.1 | -933.0 | -933.0 |
| **EXPERIMENT B** | | | | | | |
| Yield | 0.129 | | 0.070 | | 0.255 | |
| | (0.090) | | (0.144) | | (0.156) | |
| Lodging resistant | 0.447*** | 0.660*** | 0.504** | 0.601*** | 0.395 | 0.848*** |
| | (0.163) | (0.121) | (0.244) | (0.171) | (0.247) | (0.206) |
| Closed tip | 0.650** | 1.080*** | 1.046** | 1.064*** | 0.512 | 0.982*** |
| | (0.304) | (0.121) | (0.428) | (0.120) | (0.424) | (0.131) |
| Yield*Age | -0.068 | 0.401*** | -0.067 | -0.289** | -0.063 | 0.465*** |
| | (0.080) | (0.090) | (0.124) | (0.142) | (0.124) | (0.109) |
| Yield*Education | -0.026** | 0.003 | -0.034** | 0.004 | -0.008 | -0.015 |
| | (0.011) | (0.015) | (0.015) | (0.028) | (0.023) | (0.019) |
| Yield*Land under maize | -0.046 | 0.126*** | -0.014 | 0.120*** | -0.118* | 0.143*** |
| | (0.028) | (0.027) | (0.045) | (0.028) | (0.065) | (0.035) |

(*Continued*)

**Table 5.** (Continued)

| Variables | Pooled | | Male respondent | | Female respondent | |
|---|---|---|---|---|---|---|
| | Coeff | SD | Coeff | SD | Coeff | SD |
| Lodging resistant*Age | -0.191 | 0.620*** | -0.485** | 0.395 | -0.073 | 0.401 |
| | (0.146) | (0.188) | (0.213) | (0.318) | (0.218) | (0.391) |
| Lodging resistant *Education | -0.029 | 0.047* | 0.008 | 0.048 | -0.054 | 0.072 |
| | (0.020) | (0.028) | (0.027) | (0.034) | (0.037) | (0.058) |
| Lodging resistant *Land under maize | -0.003 | -0.066 | -0.083 | 0.170*** | 0.052 | 0.094 |
| | (0.042) | (0.064) | (0.069) | (0.055) | (0.067) | (0.086) |
| Lodging resistant *Region | 0.292* | 0.034 | 0.507* | -0.312 | 0.220 | 0.074 |
| | (0.165) | (0.197) | (0.278) | (0.231) | (0.231) | (0.320) |
| Closed tip*Age | 0.046** | -0.064** | -0.009 | 0.059* | 0.068* | -0.011 |
| | (0.023) | (0.026) | (0.035) | (0.033) | (0.036) | (0.055) |
| Closed tip*Education | -0.010 | 0.138** | 0.020 | 0.152*** | -0.103 | 0.218*** |
| | (0.057) | (0.070) | (0.095) | (0.052) | (0.099) | (0.073) |
| Closed tip*Land under maize | 0.173 | -0.011 | 0.007 | -0.020 | 0.398 | -0.143 |
| | (0.199) | (0.150) | (0.265) | (0.115) | (0.282) | (0.153) |
| Observations | 8,808 | 8,808 | 4,344 | 4,344 | 4,464 | 4,464 |
| chi2 | 498.6 | 498.6 | 259.2 | 259.2 | 243.4 | 243.4 |
| Log Likelihood | -2405 | -2405 | -1143 | -1143 | -1246 | -1246 |
| **EXPERIMENT C** | | | | | | |
| Yield | 0.418*** | | 0.314* | | 0.458*** | |
| | (0.104) | | (0.185) | | (0.122) | |
| Sweet taste | 1.409*** | -0.67*** | 1.370*** | -0.272 | 1.614*** | -0.621*** |
| | (0.213) | (0.174) | (0.325) | (0.305) | (0.293) | (0.179) |
| Resistant to foliar diseases | 1.133 | 2.571*** | 0.783 | 1.464*** | 3.060*** | 2.342*** |
| | (0.696) | (0.266) | (1.138) | (0.262) | (0.898) | (0.287) |
| Yield*Age | 0.187* | 0.239 | 0.017 | -0.458*** | 0.328** | -0.415*** |
| | (0.107) | (0.177) | (0.180) | (0.155) | (0.148) | (0.151) |
| Yield*Education | -0.008 | -0.002 | 0.010 | -0.030 | -0.021 | 0.022 |
| | (0.016) | (0.026) | (0.024) | (0.021) | (0.031) | (0.046) |
| Yield*Land under maize | 0.097*** | 0.127*** | 0.166*** | 0.168*** | 0.038 | 0.095*** |
| | (0.031) | (0.026) | (0.059) | (0.039) | (0.026) | (0.033) |
| Sweet taste *Age | 0.221 | 1.191*** | 0.706* | -1.409*** | 0.023 | 1.276*** |
| | (0.208) | (0.238) | (0.377) | (0.237) | (0.280) | (0.239) |
| Sweet taste *Education | 0.041 | -0.029 | 0.048 | -0.129*** | 0.212*** | 0.145** |
| | (0.033) | (0.024) | (0.043) | (0.031) | (0.075) | (0.067) |
| Sweet taste *Land under maize | 0.177** | 0.227*** | 0.281*** | -0.329*** | 0.023 | 0.062 |
| | (0.079) | (0.070) | (0.108) | (0.068) | (0.081) | (0.050) |
| Resistant to rust/pest *Age | 0.434 | 0.937** | 1.814*** | 1.904*** | 0.416 | -0.447 |
| | (0.608) | (0.473) | (0.628) | (0.360) | (0.484) | (0.310) |
| Resistant to rust/pest *Education | 0.142** | 0.109 | 0.056 | 0.207*** | 0.212* | 0.506*** |
| | (0.062) | (0.067) | (0.076) | (0.056) | (0.109) | (0.094) |
| Resistant to rust/pest *Land under maize | 0.455*** | 0.307*** | 1.078*** | 1.217*** | 0.217 | -0.144 |
| | (0.134) | (0.092) | (0.210) | (0.151) | (0.140) | (0.097) |
| Resistant to rust/pest *Region | 2.160*** | -1.425*** | 1.490* | -0.481* | 0.755 | -0.751** |
| | (0.512) | (0.214) | (0.849) | (0.257) | (0.613) | (0.293) |
| Observations | 9,264 | 9,264 | 4,680 | 4,680 | 4,584 | 4,584 |
| chi2 | 526.0 | 526.0 | 278.1 | 278.1 | 258.8 | 258.8 |

(*Continued*)

**Table 5.** (Continued)

| Variables | Pooled | | Male respondent | | Female respondent | |
|---|---|---|---|---|---|---|
| | Coeff | SD | Coeff | SD | Coeff | SD |
| Log Likelihood | -1390 | -1390 | -662.1 | -662.1 | -716.5 | -716.5 |

Standard errors in parentheses;

*** p<0.01,

** p<0.05,

* p<0.1;

SD, standard deviation;

<sup>a</sup>Region (1 = Oromia, 0 = SNNPR)

maturity trait was negatively regarded among male farmers (although not statistically significant).

During the FGDs, many farmers correctly associated early maturing varieties with low yields. Moreover, they also explained that *"if you have an early maturing crop, it is difficult to secure the crop from theft as it matures when few others in the village have a crop in the middle of the hunger season* [just before the main harvest]. *In many cases, when one has early crop, many neighbors and friends come to ask for assistance making it difficult to accumulate any yields".* They therefore preferred the higher yielding longer maturing varieties that are grown by the majority. However, we surmise that where more farmers are able to plant the short maturation variety and where market off-take is strong, these short maturing varieties are planted manly for green maize market during the short season. Given the concerns raised by farmers in these locations, and so long as market off take opportunities exist, it is possible that with better synchronization of maize planting across the villages, short season varieties may be widely planted as a source of income during the minor season (January to May). These opportunities are potentially large for farmers in peri-urban areas [49].

In Experiment B, the coefficient on the yield attribute is not significant and is lower than that observed in experiment A. This may suggest a framing effect in which the yield attribute is somewhat diminished if the variety is presented as *not lodging resistant* or has *open tip*. A variety that easily lodges may have a nugatory effect on yield such that a high yielding variety, but which is susceptible to lodging is unlikely to be chosen, not because farmers do not want high yields, but because of the risk of high losses during winds and storms. The same reasoning applies to open tip varieties which are vulnerable to ear rots because moisture easily enters the cobs. In experiment C, the relative importance of yield in relation to *sweet taste of fresh maize* or *resistance to foliar diseases* can also be seen. The coefficients for *sweet taste of fresh maize* or *resistance to foliar diseases* are larger than those for yield. Ethiopia has a large green maize market with a report indicating that by 2014, the green maize business in Addis Ababa (the largest city) was about $18 million at the time of that publication (at the exchange rate of $1 equivalent to *Birr* 18 in 2014, the *Birr* being the Ethiopian currency), thereby denoting a considerable and lucrative markets for green maize with desirable end user taste attributes [49]. Therefore, this seems to have an influence on the relative importance of taste variable in farmers mental calculations in our experiments.

## Willingness to sacrifice yield for maize variety traits between male and female farmers

Table 6 summarizes the willingness to sacrifice yield for the six maize attributes disaggregated by sex of the experiment participant. The results suggest that *early maturity* was less important

**Table 6. Mean WTSY for Ethiopia.**

|  | Pooled | Male | Female |
|---|---|---|---|
| **Experiment A** |  |  |  |
| Drought tolerant | 3.14*** | 3.26*** | 3.06*** |
| Matures in 3 months or less | -0.004 | -0.17 | 0.15 |
| **Experiment B** |  |  |  |
| Lodging resistant | 3.47*** | 7.20** | 1.55 |
| Closed tip | 5.05** | 14.96** | 2.01 |
| **Experiment C** |  |  |  |
| Sweet taste | 3.37*** | 4.36*** | 3.53*** |
| Resistant to foliar diseases | 2.71 | 2.49 | 6.69*** |

*** $p < 0.01$,

** $p < 0.05$,

* $p < 0.1$;

WTSY = willingness to sacrifice yield.

for both male and female farmers and had lowest WTSY. The early maturity trait was disliked by male participants as measured by WTSY (-0.17) and 0.15 (female farmers). In both cases, the underlying coefficient was statistically zero. *Closed tip* was valued most among male farmers (with a WTSY of 14.96) which was about twice male respondents' WTSY for *lodging resistance* (WTSY of 7.2). The WTSY for *resistance foliar diseases* was about twice the WTSY for *sweet taste* among female respondents. Among men, the WTSY for *resistance to foliar* disease was statistically zero and the WTSY for *sweet taste* was 4.4.

Comparing WTSY in experiment A and B, we see that among male farmers, *lodging resistance* was valued twice as much as *drought tolerance*. The WTSY for *sweet taste* among female respondents (3.5) was almost equal to the WTSY for *drought tolerance* (3.1) when comparing the WTSY in Experiment A and C. A similar comparison of experiment A and C shows that male farmers valued *sweet taste* (WTSY of 4.36) slightly more than *drought tolerance* (WTSY of 3.26). These results suggest that when *drought tolerance* becomes standard in many varieties, better tasting varieties will have a competitive edge, because farmers (at least in Ethiopia) seem to care about sweet taste. This is understandable as the green market is reasonably large in Ethiopia [49]. Typically, buyers appraise whether the variety is the "sweet type" by certain visible ear coloration to identify the sweet varieties.

In the aggregate, for male and female respondents, the WTSY estimates were similar in magnitude. Differences were generally small (even when they are statistically significant). This suggests that in many cases, women farmers want the same traits in maize varieties as their male counterparts. Similar results have been found by [11] in western Kenya. An important distinction is that in the Kenyan study all the respondents were participating in the study as farmers. In that role, women approach the evaluation of maize varieties in the same way as men (high yields, drought and disease tolerance and resistance to pre-harvest losses such as standability and closed tip). In their role as custodians of household food provisioning, gender differences in grain quality preferences may show more clearly. These multiple roles of women (as farmers, small scale grain retailers and custodians of family nutrition), need to be clearly studied and understood. The results also suggest that the high WTSY for *closed tip* and *lodging resistance* may be related to the high probability of losses that occur when a variety is open tip or lodges easily. In both cases, pre-harvest losses can be large. Therefore, compared to losses due to moderate mid-season drought, these occur with near certainty. Another way to

## Experiment C Session 1

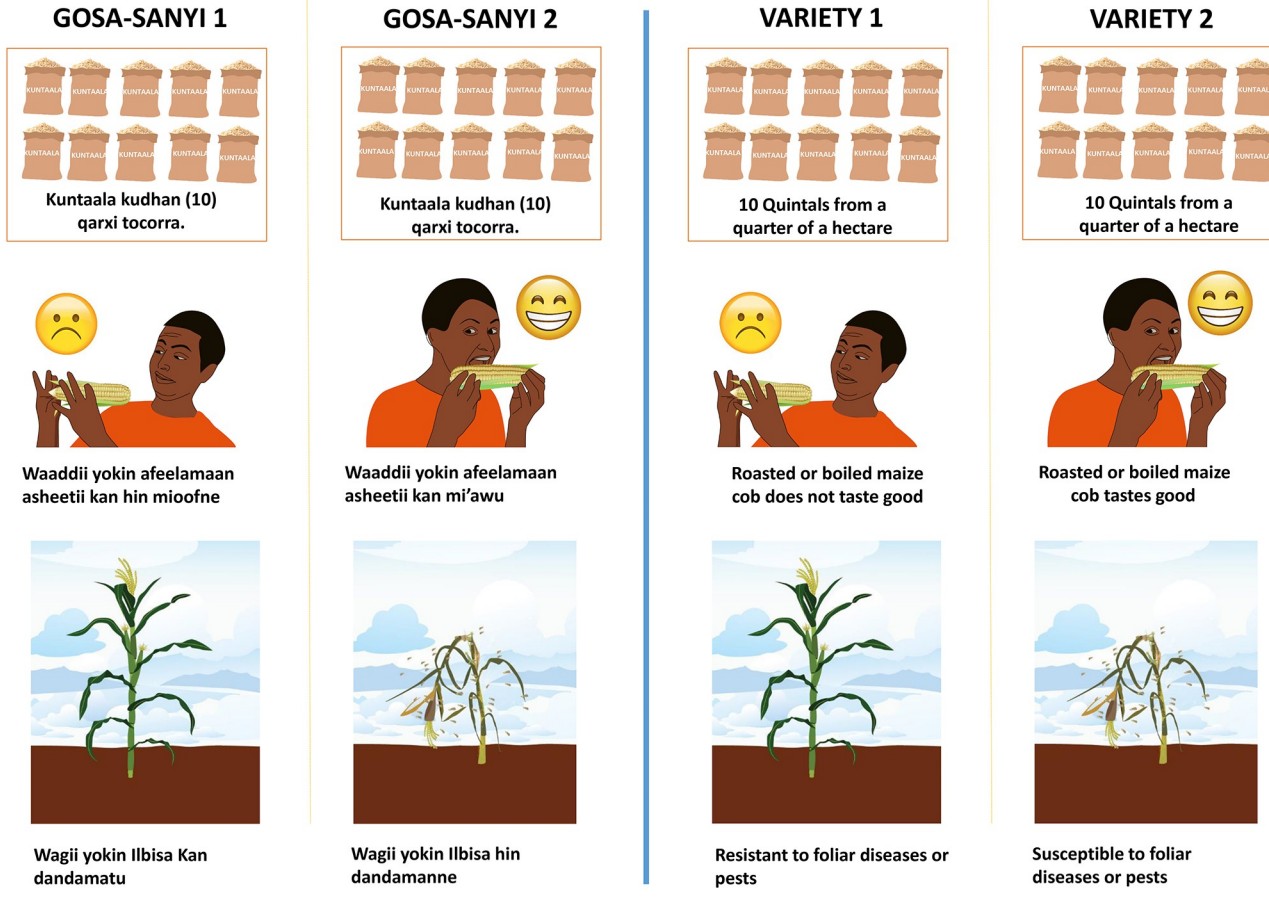

**Fig 1. Comparing willingness to pay estimates for maize traits from BDM and choice experiment.**

interpret the results is that a drought tolerant variety is considered most valuable if it is not susceptible to lodging or rotting.

The CE results from MIXL and BDM are similar in magnitude in 15 out of the 18 WTSY cases (Fig 1). This serves to provide some robustness check that the two methods (implemented on different households yielded similar results on average). The average (pooled) BDM estimate for *lodging resistance* was similar to CE pooled estimate. We conclude that even though the WTSY for *lodging resistance*, *closed tip* and *resistance to foliar diseases*, was two to five times larger in the CE estimates of WTSY., This is also true in the case of WSTY for *lodging resistance* and *closed tip* in the male sample and for *resistance to foliar diseases* for female sample. All the rest of BDM and CE estimates of WTSY were well within the same order of magnitude.

## Summary and discussions

In this paper, by applying a mixed logit model to Choice Experiment data from 1499 male and female participants, we estimated smallholder farmers' willingness to sacrifice yield for a set of critical maize traits reflecting yield potential, drought and disease tolerance, days to maturity and taste. The results suggest that in the study areas, both male and female respondents had

similar preferences for maize traits. With a WTSY of about 15, *closed tip* was valued most among male farmers. The WTSY for *resistance to foliar diseases* was about twice the WTSY for *sweet taste* among female respondents. Among male respondents, the WTSY for *resistance to foliar* disease was statistically zero and the WTSY for *sweet taste* was slightly higher than four. The WTSY for *drought tolerance* and *sweet taste* was consistently significant with an average (pooled) WTSY of 3.1 and 3.4 respectively. Overall, the WTSY as estimated by the CE and BDM methods were mostly within similar orders of magnitude. Where there were notable differences, the WTSY for lodging resistance, closed tip and resistance to foliar diseases, was two to five times larger in the CE estimates.

Broadly, in view of the reported preference patterns, the existence of multiple varieties ought to be guided by each variety bringing a unique bundle of attributes to the market. This is significant because as the seed sector grows (and becomes more competitive), many varieties will likely achieve yield parity, especially within the same market segment. This is in line with the national variety release policies and regulations in Ethiopia and most of Africa which invariably require the yields of new varieties to be significantly higher than the best commercial checks [50]. With time, as breeding efficiencies improve and breeders maintain the genetic yield gains across varieties, which is considered as a basic (or must have) trait, yield *per se* may recede as a factor of competition. At that point, market share and competition will be based on other traits such as drought tolerance or consumption traits.

A clearer view of the trends that will define future competition and product differentiation in crop improvement programs and seed system development are critical. We extend the literature on farmers' variety choices in a way that captures preference tradeoffs and that reflect farmer priorities in situations where no single variety possesses all the desired traits, as is likely to be the usual case. Such information is useful for actors along the entire maize value chain, from breeding programs to seed companies. They can use this understanding to identify market niches and segments and to adjust to new forces of competition. As part of the recent One CGIAR reform it was highlighted that the breeding programs of the organization should "ensure that new varieties are designed to meet the requirements and preferences of women and men farmers, consumers, traders, processes and others along the value chain" [51]. Along the same lines, voices from across the CGIAR and other research organizations are increasingly calling for the seed system development efforts downstream of the breeding programs to become more demand-led and inclusive [52].

In maize seed markets, farmers play a double role as consumers and producers. Therefore, they make variety choices based on a mix of consumption and market (commercial) considerations. End user traits (grain size, shape, flour conversion ratio) may, on the surface, appear inconsequential compared to yield and stress tolerance. Yet from a preference and demand perspective, these traits may well be the reason why adoption of improved varieties remain low in many countries [12, 53]. Therefore, understanding the trait trade-offs (choices) farmers are willing to make is important. To recap, no single variety can possess all the agronomic and consumption traits a farmer prefers (or requires). At some point, farmers (consumers) will be compelled to make rational trade-offs based on how they value the traits. This paper provides one of the needed analyses to improve our understanding of farmers' preferences for maize varieties in the smallholder grower segment in Ethiopia.

## Conclusions and implications

In conclusion, the results indicate that maize varieties that have potential for yield improvement should also exhibit traits that confer drought and disease tolerance and where the market is sensitive, they should also satisfy end user consumption tastes. In areas where green maize

markets are developed, the yield advantage of less tasty varieties may have to be quite substantial to win market share, especially if taste-sensitive green maize markets are strong. Extra early varieties are not necessarily preferred given their perceived low yields (at least in this environment) where locally adapted, higher yielding, medium (110–120-day) maturity hybrids are available. Given the high risks of pre-harvest losses from open tip and easily-lodging varieties, these varieties should not be advanced in any breeding pipelines meant for grain harvesting.

We therefore suggest that when varieties in a particular market attain yield parity and drought tolerant (and other stress-tolerant) maize varieties become widely available, better tasting varieties will be preferred within the class of high-yielding and stress-tolerant varieties. Where drought challenges are not severe or only of a minor concern, such as in the humid highlands, it is conceivable that some farmers in those environments may be willing to forgo drought tolerant varieties in favor of sweet taste, where markets demand such. In mid-altitude sub-humid areas, where mid-season droughts may be a major production threat, drought tolerance traits will likely dominate in farmers' variety choices. Breeding programs could therefore consider approaches that carefully combine various desired attributes (such as taste, tolerance to pre-harvest losses and disease resistance) while enhancing drought tolerance and yield potential in varietal development.

The study's main aim was to capture heterogeneity in preferences. While it is not possible to segment the market endlessly, the final and realistic segmentation will depend on the commercial viability or social impact potential of each segment. The weakness of the current seed markets in Ethiopia and the region is that the final seed products are not well-differentiated at the last-mile marketing and distribution level. Finally, we note that our study may not provide the full range of social and economic factors that drive variety choices. Future studies should focus on a deeper dive qualitative analyses to supplement quantitative approaches. These qualitative studies will need to highlight the social and economic forces not captured in econometric equations or which are important but may not be empirically testable.

## Supporting information

**S1 File. This file has three sections on the CE, BDM and supplementary regressions.**
(DOCX)

**S1 Data. This file has the supporting data used for the analysis.**
(ZIP)

## Author Contributions

**Conceptualization:** Paswel Marenya.

**Data curation:** Paswel Marenya.

**Formal analysis:** Paswel Marenya, Rosina Wanyama, Solomon Alemu.

**Funding acquisition:** Paswel Marenya, Ola Westengen.

**Investigation:** Paswel Marenya, Solomon Alemu.

**Methodology:** Paswel Marenya.

**Project administration:** Paswel Marenya, Solomon Alemu, Ola Westengen.

**Supervision:** Paswel Marenya.

**Validation:** Paswel Marenya, Ola Westengen.

**Writing – original draft:** Paswel Marenya, Rosina Wanyama.

**Writing – review & editing:** Paswel Marenya, Ola Westengen, Moti Jaleta.

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
