## [Decision Letter · Decision Letter 0]

23 Jun 2022

PONE-D-22-06010Maize variety preferences among smallholder farmers in Ethiopia: implications for demand-led breeding and seed system developmentPLOS ONE

Dear Dr. Marenya,

Thank you for submitting your manuscript to PLOS ONE. After careful consideration, we feel that it has merit but does not fully meet PLOS ONE’s publication criteria as it currently stands. Therefore, we invite you to submit a revised version of the manuscript that addresses the points raised during the review process.

Based reviewers' comments: Major Revision.

We look forward to receiving your revised manuscript.

Kind regards,

Guangyuan He, PhD

Academic Editor

PLOS ONE

Journal Requirements:

2. We note that Figure (1) in your submission contain [map/satellite] images which may be copyrighted. All PLOS content is published under the Creative Commons Attribution License (CC BY 4.0), which means that the manuscript, images, and Supporting Information files will be freely available online, and any third party is permitted to access, download, copy, distribute, and use these materials in any way, even commercially, with proper attribution. For these reasons, we cannot publish previously copyrighted maps or satellite images created using proprietary data, such as Google software (Google Maps, Street View, and Earth). For more information, see our copyright guidelines: http://journals.plos.org/plosone/s/licenses-and-copyright.

1. You may seek permission from the original copyright holder of Figure (1) to publish the content specifically under the CC BY 4.0 license.  

Reviewers' comments:

Reviewer's Responses to Questions

**Comments to the Author**

1. Is the manuscript technically sound, and do the data support the conclusions?

Reviewer #1: Yes

Reviewer #2: Partly

2. Has the statistical analysis been performed appropriately and rigorously? 

Reviewer #1: Yes

Reviewer #2: N/A

3. Have the authors made all data underlying the findings in their manuscript fully available?

Reviewer #1: No

Reviewer #2: No

4. Is the manuscript presented in an intelligible fashion and written in standard English?

Reviewer #1: Yes

Reviewer #2: Yes

5. Review Comments to the Author

Reviewer #1: It is a well-written paper with relevant implications for germplasm development and the seed market system. It is interesting to understand framers' yield trade-off because of their choice of other attributes. The use of a combination of methods for robustness is also commendable.

Comments

The recommendation breeding program and seed system should consider heterogeneity, especially gender difference, to develop a gender-inclusive market system is tricky for this paper. Almost all the sampled females are wives within the male-headed households. How practical and economical is this recommendation to countries with limited resources and diverse development priorities to respond to the preferences of two individuals living under the same roof. Let's assume this is possible, does the household have adequate land and other resources to accommodate this. It would have been interesting if you could have also interviewed both spouses together and analyzed their joint decision-making process. There could be a knowledge gap between the two, but jointly they can provide answers close to actual. This could have also avoided potential result bias because of drifting and difficulty understanding and following up a series of questions*. Also, closely looking at the summary statistics, the land size is more or less the same across sample-that means there is little variation within the community. Variables can turn into significance by chance. I suggest the authors show results with and without the interaction of variables.

The challenge with using the CE and BDM approaches is that less-educated respondents may not easily comprehend and follow the number of experiments/sessions. I wonder if the authors have used a validation mechanism that shows farmers' ability to understand the entire process.

You seem to have more than one observation per unit? Why not apply the panel data model.

Under table 5, what type of standard errors were reported? Robust or ….

Translate Figure 1 into English. Someone might want to replicate this study. In the data section, indicate questions were translated to the local language.

Reviewer #2: In this paper, the demand of farmers and markets for local variety characteristics in different regions was understood through farmer questionnaire, so as to provide reference for making future breeding plans. Wishes are good.

There are some problems with this kind of survey report. What does it mean that men and women respond differently? How significant is it? Some results of statistical analysis have statistically significant differences, which may not have practical or scientific significance. Statistical analysis must have been explicable in science.

What is the direct basis on which farmers make their choices? It should be years of experience and demand for local suitable varieties. Ultimately, it is decided by local climate conditions, cultivation conditions, and people's demand for high yield and quality. Therefore, on the basis of the above questionnaire, the following contents should be suggested in order to provide valuable reference for the establishment of breeding programs for future.

1 .Climatic conditions (water, temperature, soil conditions), crop rotation system, cultivation measures, people's preferences (summary of questionnaire results).

2. the yield, growth period, nutritional quality, taste quality, processing quality, lodging resistance, diseases, pests and other traits of varieties. Comparison of varieties between different regions and different ages, pointing out the characteristics.

3. According to the above results, the cultivation regions and variety types could be divided or revised.

4. Different breeding objectives could be proposed for different regions.

6. PLOS authors have the option to publish the peer review history of their article (what does this mean?). If published, this will include your full peer review and any attached files.

Reviewer #1: **Yes: **Menale Kassie

Reviewer #2: No

---

## [Author Response · Author response to Decision Letter 0]

18 Aug 2022

Introduction

Below we respond to the reviewers’ comments and document where the necessary changes have been made in the manuscript

Reviewer #1: 

It is a well-written paper with relevant implications for germplasm development and the seed market system. It is interesting to understand framers' yield trade-off because of their choice of other attributes. The use of a combination of methods for robustness is also commendable

Response: We thank the reviewer for this compliment and for taking time to read and offer suggestions for improvement. We have clarified the issues raised as stated in the following replies. 

Comments

The recommendation breeding program and seed system should consider heterogeneity, especially gender difference, to develop a gender-inclusive market system is tricky for this paper. Almost all the sampled females are wives within the male-headed households. 

• How practical and economical is this recommendation to countries with limited resources and diverse development priorities to respond to the preferences of two individuals living under the same roof. Let's assume this is possible, does the household have adequate land and other resources to accommodate this. 

Response: We do not recommend that breeding institutions develop separate breeding programs for men and women separately. We have clarified that our study was to capture heterogeneity of preferences. One way to do this in a rural agrarian setting is to elicit the views of men and women (the most important binary demographic category). Our study doesn’t find dramatic differences between men and women. Where they exist these differences tend to be small. We then control for the other demographic factors as explained in the methodology section. We have clarified this matter and our last concluding sentence now reads as follows:

The study’s main aim was to capture heterogeneity in preferences. While it is not 

possible to segment the market endlessly, the final and realistic segmentation will depend on the commercial viability or social impact potential of each segment.

• It would have been interesting if you could have also interviewed both spouses together and analyzed their joint decision-making process. There could be a knowledge gap between the two, but jointly they can provide answers close to actual. This could have also avoided potential result bias because of drifting and difficulty understanding and following up a series of questions*

Response: We are aware that some authors have used the approach suggested by the reviewer. In our judgement, interviewing both spouses separately may lead to less biased answers as each individual is free to state their opinion without trying to conform to the answers of the spouse or another family member. Please note that to reduce cognitive burden, we used picture in the local language (see fig 2). Also, the interviewers spoke the local language. 

• Also, closely looking at the summary statistics, the land size is more or less the same across sample-that means there is little variation within the community. Variables can turn into significance by chance. I suggest the authors show results with and without the interaction of variables.

• The challenge with using the CE and BDM approaches is that less-educated respondents may not easily comprehend and follow the number of experiments/sessions. I wonder if the authors have used a validation mechanism that shows farmers' ability to understand the entire process.

Response: Thanks for the observation. In terms of presenting estimates that were not interacting, we believe Table 5 is sufficient, and that table already has the estimates for each trait without interaction. Recall the decision variables are the traits (the respondents were choosing the trait combinations) and the experiment was designed detect the willingness to pay for the traits. The demographic variables only come through in terms of how they interact with the traits (the decision variables). Nevertheless, we have added the estimates Appendix 3 (Table A3).

In terms of comprehension, we explain how we ensured adequate comprehension and how we reduced the cognitive burden by conducting the experiment in local language. We now ensure this additional information is in the paper (Appendix 1).

To test whether the respondents truly understood the question and before making a final selection in any experimental session, the interviewers were instructed to allow the respondent a minimum of 20-30 seconds to evaluate the choice before settling on the preferred alternative. More time was allowed for the respondent to ask questions. The combination of un-rushed choice process and the chance to ask clarifying questions, preliminary explanation in each session and use of vivid pictures ensured the there was low cognitive burden on the respondents and comprehension was near-guaranteed in almost all cases. 

• You seem to have more than one observation per unit? Why not apply the panel data model.

Under table 5, what type of standard errors were reported? Robust or ….

Response: The reviewer raises a good point. It is possible that the responses in Equation (that is Y) are correlated within households. In our case the MIXL estimation was clustered at household level, and the standard errors are specified as robust. The MIXL already has an inbuild algorithm that takes account of this via a group variable in STATA implementation. 1The mixed logit (MIXL) which is meant to take care of heterogeneity uses a group variable that allows the model to be estimated by clustering at the household respondent and household level. We used the variable “group_id” which was generated using three variables household ID (household cluster), respondent type (respondent heterogeneity) and experiment session id (to capture any correlations between experimental session. This issue is now explained in Footnote #5 in the manuscript (pasted below):

It can be expected that in Equation 9 the responses are correlated which could mean that the intra-household responses are not independent. This is a fair assumption given that the respondents are part of the same household. To handle this, we implemented the MIXL in STATA 16. The estimation in this procedure was clustered at household level with robust standard. Note that the multiple observations within the household has a “panel” structure. In our case, the MIXL is designed to take this into account via and sub-routine “group_id” during MIXL implementation. The “group_id” was meant to identify the cluster the respondents through the household identifier (household cluster), respondent type (intra-household respondent heterogeneity) and experiment session number (to capture any correlations between experimental session.

• Translate Figure 1 into English. Someone might want to replicate this study. In the data section, indicate questions were translated to the local language.

Response: This is a good point. We have added a translated picture with English translations.

---

## [Decision Letter · Decision Letter 1]

25 Aug 2022

Maize variety preferences among smallholder farmers in Ethiopia: implications for demand-led breeding and seed sector development

PONE-D-22-06010R1

Dear Dr. Marenya,

We’re pleased to inform you that your manuscript has been judged scientifically suitable for publication and will be formally accepted for publication once it meets all outstanding technical requirements.

Kind regards,

Guangyuan He, PhD

Academic Editor

PLOS ONE

Additional Editor Comments (optional):

Reviewers' comments:

Reviewer's Responses to Questions

**Comments to the Author**

1. If the authors have adequately addressed your comments raised in a previous round of review and you feel that this manuscript is now acceptable for publication, you may indicate that here to bypass the “Comments to the Author” section, enter your conflict of interest statement in the “Confidential to Editor” section, and submit your "Accept" recommendation.

Reviewer #1: All comments have been addressed

Reviewer #2: All comments have been addressed

2. Is the manuscript technically sound, and do the data support the conclusions?

Reviewer #1: Yes

Reviewer #2: Yes

3. Has the statistical analysis been performed appropriately and rigorously? 

Reviewer #1: Yes

Reviewer #2: Yes

4. Have the authors made all data underlying the findings in their manuscript fully available?

Reviewer #1: No

Reviewer #2: Yes

5. Is the manuscript presented in an intelligible fashion and written in standard English?

Reviewer #1: Yes

Reviewer #2: Yes

6. Review Comments to the Author

Reviewer #1: The paper employs sound statistical analysis and is written using good English. However, I couldn't see if the Authors had submitted the data used for analysis.

Reviewer #2: (No Response)

7. PLOS authors have the option to publish the peer review history of their article (what does this mean?). If published, this will include your full peer review and any attached files.

Reviewer #1: **Yes: **Menale Kassie

Reviewer #2: No

---

## [Editor Report · Acceptance letter]

7 Sep 2022

PONE-D-22-06010R1 

Maize variety preferences among smallholder farmers in Ethiopia: implications for demand-led breeding and seed sector development 

Dear Dr. Marenya:

I'm pleased to inform you that your manuscript has been deemed suitable for publication in PLOS ONE. Congratulations! Your manuscript is now with our production department. 

Kind regards, 

on behalf of

Prof. Guangyuan He 

Academic Editor

PLOS ONE